# Landscape of Genomic Alterations and PD-L1 Expression in Early-Stage Non-Small-Cell Lung Cancer (NSCLC)—A Single Center, Retrospective Observational Study

**DOI:** 10.3390/ijms232012511

**Published:** 2022-10-19

**Authors:** Susann Stephan-Falkenau, Anna Streubel, Thomas Mairinger, Jens Kollmeier, Daniel Misch, Sebastian Thiel, Torsten Bauer, Joachim Pfannschmidt, Manuel Hollmann, Michael Wessolly, Torsten Gerriet Blum

**Affiliations:** 1Institute for Tissue Diagnostics, MVZ at Helios Klinikum Emil von Behring, 14165 Berlin, Germany; 2Department of Pneumology, Heckeshorn Lung Clinic, Helios Klinikum Emil von Behring, 14165 Berlin, Germany; 3Department of Thoracic Surgery, Heckeshorn Lung Clinic, Helios Klinikum Emil von Behring, 14165 Berlin, Germany; 4Institute of Pathology, University Hospital Essen, University of Duisburg-Essen, 45147 Essen, Germany; 5German Cancer Consortium (DKTK), Partner Site University Hospital Essen, 45147 Essen, Germany

**Keywords:** lung cancer, molecular pathology, precision oncology, biomarker-driven targeted therapy, early-stage NSCLC, EGFR mutation, adjuvant TKI therapy, PD-L1 expression, immunotherapy

## Abstract

Precision oncology and immunotherapy have revolutionized the treatment of advanced non-small-cell lung cancer (NSCLC). Emerging studies show that targeted therapies are also beneficial for patients with driver alterations such as epidermal growth factor receptor (EGFR) mutations in early-stage NSCLC (stages I–IIIA). Furthermore, patients with elevated programmed death-ligand 1 (PD-L1) expression appear to respond favorably to adjuvant immunotherapy. To determine the frequency of genomic alterations and PD-L1 status in early-stage NSCLC, we retrospectively analyzed data from 2066 unselected, single-center patients with NSCLC diagnosed using next-generation sequencing and immunohistochemistry. Nine-hundred and sixty-two patients (46.9%) presented with early-stage NSCLC. Of these, 37.0% had genomic alterations for which targeted therapies have already been approved for advanced NSCLC. The frequencies of driver mutations in the early stages were equivalent to those in advanced stages, i.e., the rates of EGFR mutations in adenocarcinomas were 12.7% (72/567) and 12.0% (78/650) in early and advanced NSCLC, respectively (*p* = 0778). In addition, 46.3% of early-stage NSCLC cases were PD-L1-positive, with a tumor proportion score (TPS) of ≥1%. With comparable frequencies of driver mutations in early and advanced NSCLC and PD-L1 overexpression in nearly half of patients with early-stage NSCLC, a broad spectrum of biomarkers for adjuvant and neoadjuvant therapies is available, and several are currently being investigated in clinical trials.

## 1. Introduction

The recommended treatment of choice for patients with early-stage NSCLC (stages I–IIIA) is complete surgical resection [1,2]. Despite the curative approach, the 5-year overall survival (OS) after surgical NSCLC resection is stage dependent, decreasing from 92% in stage IA1 to 36% in stage IIIA [3]. Adjuvant chemotherapy contributes to a modest 5-year OS benefit of 4–5% [4,5]. Consequently, with the advancing technological capabilities in molecular testing and the continued exploration of prognostic and predictive biomarkers, novel therapeutic strategies are currently under investigation. Recent data have shown that curatively resected EGFR-mutation-positive NSCLC patients in stages IB–IIIA may benefit from adjuvant therapy with tyrosine kinase inhibitors (TKIs) [6,7,8]. In the pivotal ADAURA trial, adjuvant therapy with the third-generation EGFR-TKI osimertinib significantly prolonged disease-free survival (DFS) of EGFR-mutant NSCLC patients compared with the placebo. In the overall population (patients with stage IB–IIIA disease), 89% of the patients in the osimertinib group (95% CI, 85 to 92) and 52% of those in the placebo group (95% CI, 46 to 58) were alive and disease-free at 24 months (overall hazard ratio for disease recurrence or death, 0.20; 99.12% CI, 0.14 to 0.30; *p* < 0.001) [9]. Likewise, in the neoadjuvant setting, several studies have shown good responses to TKI therapies, resulting in pathology down-staging and restoration of operability [10,11,12,13,14,15].

Treatment with TKIs in EGFR mutation-positive patients with advanced NSCLC (stages IIIB–IV) is a long-accepted standard [1,2]. Numerous studies have characterized and categorized different EGFR mutations in terms of their individual prognoses and response to TKI therapy [16,17,18]. In order to determine the best personalized therapies in NSCLC patients, professionals need to know exactly what specific subtype of mutation is present (i.e., common, uncommon, and compound EGFR mutations). Furthermore, the co-occurrence of other mutations in addition to an EGFR mutation also appears to influence prognosis and response to treatment [19,20,21,22]. In addition to TKI-sensitive EGFR mutations, other activating molecular alterations, such as ALK translocations, are examined in current trials regarding their suitability as targets for adjuvant or neoadjuvant therapeutic approaches. The approvals of some of these targeted drugs are very likely in the coming years [23,24,25,26]. 

Immunotherapies, namely, checkpoint inhibitors (CPIs), represent the second mainstay of current cancer treatment for patients with advanced NSCLC. Equally, their utility in the adjuvant and neoadjuvant settings has gained increasing interest more recently. In contrast to advanced tumor stages, in which CPIs are the current gold standard for patients without targetable molecular alterations [1,2], therapeutic options are limited in resectable early-stage NSCLC in the absence of genetic driver alterations. In patients with resected stage II–IIIA NSCLC, atezolizumab is the only approved CPI to date, based on the positive results of a phase III trial (IMpower010), demonstrating a DFS benefit compared to the best supportive care: a 34% reduced risk of tumor recurrence or death in the subgroup whose tumors expressed PD-L1 (HR: 0.66; 95% Cl: 0.50; 0.88; *p* = 0.0039) [27]. The results of several ongoing trials in early stage NSCLC patients evaluating the efficacy of immunotherapies in the adjuvant and neoadjuvant setting are eagerly awaited and may expand the choices of CPIs for this indication [28,29,30,31,32,33].

Generally, questions arise as to whether the well-established findings on targeted and immunotherapies in advanced-stage NSCLC can be transferred to adjuvant and neoadjuvant therapies. The answer also implies the need for comprehensive molecular analysis by high-throughput techniques regardless of tumor stage as opposed to single-gene testing. The latter is currently often limited to screening for exon 19 deletions and L858R substitution in EGFR-mutated patients undergoing surgery due to proof of eligibility for adjuvant osimertinib therapy in completely resected early-stage NSCLC patients [34]. 

As much as we know about the genomic landscape and antitumor immune response in advanced-stage NSCLC, little has been published on the prevalence and nature of EGFR mutations and other driver alterations and PD-L1 expression in early-stage NSCLC [22,35,36,37,38,39,40]. Therefore, in our study, we investigated the frequencies of genomic alterations and PD-L1 status in patients with primary diagnosed stage I–IIIA NSCLC to elucidate potential biomarkers for precision oncology and to explore the importance of broad-range sequencing in early tumor stages.

## 2. Results

### 2.1. Patient Characteristics

Two-thousand and sixty-six patients were diagnosed with NSCLC at the Lung Cancer Centre, Lungenklinik Heckeshorn, Helios Klinikum Emil von Behring (HKEvB), Berlin, Germany, between 04/2017 and 02/2021. All patients were of Caucasian descent. A complete TNM stage was determined in 2052 (99.3%) patients. In total, 962 patients (46.9%) presented with early-stage NSCLC (stages I–IIIA) and 1090 patients (53.1%) with advanced stage IIIB–IV NSCLC disease at the time of diagnosis. Tumor tissue from 1550 patients (75.5%) was eligible for DNA- and RNA-based molecular analyses, including 723 patients (46.6%) with early-stage and 827 (53.4%) with advanced-stage disease. For the early stages, the tumor collection site was available in 98.3% (711/723) of patients; the tumor-confirming tissue originated from the lungs and loco-regional metastases in 92.3% (656/711) and 7.7% (55/711), respectively. For the advanced stages, the tumor collection site was available in 98.2% (812/827) of patients; tumor tissue was obtained in 60.2% (489/812) of patients from the primary pulmonary tumor, in 27.0% (219/812) from loco-regional metastases, and in 12.8% (104/812) from distant metastases (pleura, bone, liver, brain, skin). The histological subtypes of early and advanced-stage NSCLC encompassed adenocarcinomas (ACAs), squamous cell carcinomas (SCCs), adenosquamous carcinomas (ASCs), sarcomatoid carcinomas (SCs), large cell carcinomas (LCCs), large cell neuroendocrine carcinomas (LNECs), other histological NSCLC subtypes (other NSCLC), and NSCLC that could not be otherwise specified (NSCLC NOS). Baseline patient and tumor characteristics, including the distributions of histological subtypes for early and advanced-stage disease, are listed in Table 1. Except for grading, no statistically significant differences were seen between the groups. Of note, however, 5.7% (41/723) of early-stage and 1.2% (10/827) of advanced-stage patients presented with two or three primary lung cancers metachronously or synchronously within the study period (*p* < 0.001).

### 2.2. Overall Frequency of Genomic Alterations in Early-Stage Non-Small-Cell Lung Cancer 

Of all the patients with early-stage NSCLC with molecular testing, 86.7% (627/723) showed one or more genomic alteration. We found that 13.3% (96/723) of patients had no detectable mutations or fusions in the analyzed genes. In the advanced tumor stages, IIIB–IV, genomic alterations were slightly more frequent. There was one or more detected genomic alteration in 90.6% (749/827), and no mutations or gene fusions were seen in 9.4% (78/827). Next, 37.0% (210/567) of patients with primary diagnosed ACA in the early stages, I–IIIA, displayed genomic driver alterations that could be targeted by TKI therapies currently approved in advanced NSCLC (EGFR, KRAS G12C, BRAF V600E, ERBB2, ALK, ROS1, METEx14skip, RET). Another 48.5% (275/567) showed at least one or more mutation or a gene rearrangement in other genes. Only 14.5% (82/567) had no alterations in the genes examined, as shown in Figure 1.

There was no significant difference in the mutation rate in ACAs obtained from primary pulmonary tumors compared with ACA metastases in loco-regional lymph-node tissue. In the early-stage ACAs, the tumor collection site was available in 558 patients (558/567, 98.4%); the tumor-confirming tissue originated from the lungs and loco-regional metastases in 92.1% (514/558) and 7.9% (44/558) of patients, respectively. However, the results should be interpreted with caution, particularly for low mutation rates, because of the small number of loco-regional lymph node metastases studied (Table 2).

Patients with ACA in advanced stages had actionable genomic alterations in 37.3% of cases (242/650). Another 51.9% (337/650) had at least one or more mutation or a gene rearrangement in relevant genes, and only 10.8% (71/650) of patients had no alterations detected in the genes studied (Table 3).

Detailed information on the mutation frequency in pulmonary adenocarcinoma according to the early tumor stages IA1, IA2, IA3, IB, II, and IIIA is presented in Appendix A.

Tumors with non-ACA histology were found in 21.6% (156/723) of the tested early-stage NSCLC and in 21.4% (177/650) of the advanced-stage NSCLC. In both groups, significantly lower frequencies of currently actionable driver mutations than in ACAs were detected: 10.9% (17/156) vs. 37.0% (210/567) in early stages (*p* < 0.001) and 14.7% (26/177) vs. 37.3% (242/650) in advanced stages (*p* < 0.001).

### 2.3. EGFR

#### 2.3.1. Frequency and Classification

The frequency of EGFR mutations in early-stage ACAs was 12.7% (72/567). Only two tumors with non-ACA histology harboring EGFR mutations, namely, one SCC (1.2%; 1/84; p.Gly719Ala; stage IIIA) and one SC (5.3%; 1/19; p.Leu858Arg, stage IIIA), were detected. Firstly, 72.2% of the EGFR mutations detected in ACAs were common mutations—exon 19 deletions or L858R substitutions (45.8%, 33/72 resp. 26.4%, 19/72). Secondly, 19.4% (14/72) were uncommon EGFR mutations, and 8.3% (6/72) of patients presented with compound mutations, as shown in Figure 2. According to the classification of uncommon EGFR mutations followed by Yang et al. [41], we assigned 9.7% (7/72) mutations as class I (point mutations in exon 18, 19, 20, and 21), no mutations as class II (primary p.Thr790Met), and 9.7% (7/72) mutations as class III (exon 20 insertions). However, following a recently published new classification of rare EGFR mutations by Janning et al. [42], we found 55.0% (11/20) group 1 mutations (G719X, S7681, L861Q, and combinations) that enable higher efficacy of EGFR-TKI in comparison to chemotherapy. Another 35.0% (7/20) were classified as group 2 mutations (exon 20 insertions), most of them not TKI responsive. In addition, 10.0% (2/20) corresponded to group 3 mutations (very rare point mutations and complex EGFR mutations containing exon 19 deletions or L858R mutations). In advanced tumor stages, IIIB–IV, we found EGFR mutations in 12.0% (78/650) of ACAs, in 1.1% of SCCs (1/87; p.Glu865Lys), and in 6.7% of ASCs (1/15; p.Ser768_Asp770dupSerValAsp). In total, 69.2% of EGFR-mutated ACAs had common mutations, including 41.0% exon 19 deletions (32/78) and 28.2% Leu858Arg (22/78). Another 19.2% (15/78) of ACAs had uncommon mutations: 10.3% class I (8/78), no class II, and 9.0% class III (7/78) mutations. The frequency of compound mutations was 11.5% (9/78); see Figure 2. A list of all detected mutations is presented in Appendix A.

#### 2.3.2. Mutations Co-Occurring in Early-Stage NSCLC with EGFR Mutations

In addition to EGFR driver mutations, we found co-occurring mutations in ACA less frequently in early compared to advanced stages: 52.8% (38/72) and 73.1% (57/78), respectively (*p* = 0.016) (Figure 3). Most frequently, missense mutations in the tumor suppressor gene TP53 were detected (34.7% in early-stage versus 46.2% in advanced-stage tumors, *p*-value: 0.208). Of note, among patients with exon 19 deletion, 6.9% (5/72) had an additional TP53 mutation in exon 8. In addition to TP53 mutations, we found functional mutations in PIK3CA (2.8% vs. 6.4%, *p*-value 0.505), CTNNB 1 (2.8% vs. 3.8%, *p* = 1), SMAD4 (5.6% vs. 1.3%, *p* = 0.1950), STK11 (1.4% vs. 1.3%, *p* = 1), and BRAF nonV600 (1.4% vs. 1.3%, *p* = 1). Mutations in PTEN, FGFR3, ERBB2, ERBB4, and FBXW7 occurred exclusively in the advanced stages (Table 2). Among EGFR-mutated tumors with non-ACA histology, only the SC in stage IIIA had an activating mutation in PIK3CA (p.E545K).

### 2.4. KRAS

In total, 40.4% (229/567) of patients with early-stage ACA carried a KRAS mutation. The mutations were distributed among exons as follows: 95.2% exon 2, 4.4% exon 3, and 0.4% exon 4. One patient had two different KRAS mutations simultaneously (p.Gly13Asp and p.Thr148Arg). Note that 42.8% (98/229) of all KRAS-mutated patients had a targetable p.G12C mutation. A detailed list of all detected KRAS mutations is provided in Appendix A. Interestingly, 34.7% (34/98) of patients had the concomitant presence of one mutation each in KRAS p.G12C and TP53. Another 9.6% (22/229) of all patients with activating KRAS mutations had a co-occurring mutation in STK11. In patients with non-ACA histology, KRAS mutations were found in only 7.6% (12/156) of early-stage cases (92.9% exon 2, 7.1% exon 3.0% exon 4), of which 50% (6/12) were p.G12C. Patients with advanced-stage ACA had a KRAS mutation 40% (270/650) of the time (92.3% exon 2, 7.7% exon 3, 0% exon 4). Four patients had two different KRAS mutations simultaneously (2 × p.Gly12Val and p.Gln61Leu; 2 × p.Gly12Val and P.Gln61His). Further, 39.2% (102/270) of KRAS-mutated patients presented with KRAS P.G12C. In 38.2% (39/102) of these patients, a co-occurring mutation in TP53 was detected. Additionally, 8.5% (23/270) of all patients with activating KRAS mutations also had a mutation in STK11. Advanced-stage patients with a non-ACA histology had an activating KRAS mutation 15.8% (28/177) of the time (92.9% exon 2, 7.1% exon 3, 0% exon 4); 43.5% (10/23) of these patients had a P.G12C mutation. 

### 2.5. BRAF

Note that 5.6% (32/567) of patients with early-stage ACA had a BRAF mutation; among them, 31.3% (10/32) had the targetable p.V600E mutation. Of the remaining patients with a non-p.V600E mutation, three had a co-occurring activating KRAS mutation (2 × p.G12C, 1 × p.G12V). In comparison, 5.5% (36/650) of patients with advanced-stage ACA had a BRAF mutation; for 38.9% (14/36) of them, p.V600E. The remaining patients with a non-p.V600E mutation exhibited an additional and concomitant activating KRAS mutation in three cases (p.G12C, p.G12D, p.G13C). Interestingly, one patient with ACA presented with a TRIM24–BRAF (T12B10) fusion. Patients with non-ACA histology showed one BRAF mutation (p.G469A) in early-stage (0.64%, 1/156) and six BRAF mutations in advanced-stage disease (3.4%, 6/177), including two p.V600E and one non-p.V600E harboring an additional KRAS mutation (p.G12C). TP53 mutations were found in addition to the p.V600E mutation in five patients with early ACA (50%, 5/10) and in six patients with advanced ACA (42.8%, 6/14).

### 2.6. ERBB2 (HER2)

Overall, patients with early-stage ACA displayed activating mutations in ERBB2 0.88% (5/567) of the time; 60% (3/5) presented an exon 20 insertion. The remaining two patients showed a single-base substitution in exon 19 (p.L755P and p.I767M). Patients with advanced-stage ACA had an activating mutation in ERBB2 in 1.5% (10/650) of cases; exon 20 insertions were found in 63.6% (7/10), along with single-base substitutions in exon 19 (p.L755S), exon 8 (p.S310F), and exon 21 (p.V859D). One patient had two mutations in the ERBB2 gene at the same time (p.S310F + p.Val777delinsValGlySerPro). In addition, there was one patient with an exon 20 insertion in ERBB2 and an activating mutation in the EGFR gene p.L861Q. TP53 co-mutations occurred in both early and advanced stages in two patients each, including one exon 20 insertion and a one-point mutation each in exon 19 and 21.

### 2.7. ALK

RNA-based fusion analysis detected ALK rearrangements in a total of 1.2% (7/567) of patients with early-stage ACA. In 6/7 patients, the fusion partner was found to be EML4. Variant 1 (E13; A20) was present in 50.0% (3/6), variant 2 (E20; A20) in 33.3% (2/6), and variant 5a (E2; A20) in 16.7% (1/6). One patient presented with an ALK–KIF5B fusion. Variant 3 (E6a/b; A20) was not present. None of the patients had a co-occurring mutation. Only one early-stage patient with non-ACA histology (carcinosarcoma) had an EML4–ALK fusion (variant 1, E13; A20). In the advanced tumor stages, patients with ACA had ALK fusions with 2.2% frequency (14/650), and the fusion partner was always EML4. Next, 42.8% (6/14) had variant 1, 7.1% (1/14) variant 2, 42.8% (6/14) variant 3 (E6a/b; A20), and 7.1% (1/14) variant 5. Five of the fourteen patients (35.7%) had a co-occurring mutation in TP53. Of the patients with variant 3 (E6a; A20), one patient (16.7%, 1/6) presented with a co-occurring mutation in TP53 (p.F134I). Only one patient with non-ACA histology (LNEC) harbored an EML4–ALK fusion (variant 3, E6a; A20); an associated TP53 mutation was not present.

### 2.8. ROS1

ROS1 rearrangement could be detected in the early-stage ACA group in only one patient (0.2%, 1/576) as an SDC4–ROS1 fusion. In advanced tumor stages, ACA showed ROS1 fusions in 0.6% (4/650) of cases: two patients with EZR-ROS1, one patient with SDC 4–ROS1, and one patient with CD74–ROS1 fusion. We found no concomitant passenger mutation in any of the cases. No ROS1 fusions were found in the patients with non-ACA histology.

### 2.9. RET

RET fusions were seen in 0.7% (4/567) of patients with early-stage ACA, including three KIF5B–RET fusions and one CCDC6–RET fusion. TP53 co-mutation did not occur. In contrast, no RET fusions were found in tumors with non-ACA histology. In the advanced stages, 0.5% (3/650) of patients with ACA had a RET fusion (2 × KIF5B-RET, 1 × CCDC6-RET), and one patient with a KIF5B–RET fusion additionally presented with a co-occurring TP53 mutation. Among patients with non-ACA histology, one SCC harbored a KIF5B–RET fusion (0.6%, 1/650).

### 2.10. MET EX14 Skipping Events

RNA-based fusion analysis revealed a MET exon 14 skipping event in 2.3% (13/567) of patients with early-stage ACA. Correspondingly, genomic mutations were detected mainly in the splicing region of exon 14, both in the 5’ and 3’ regions. Next, 38.5% (5/13) of tumors showed a concomitant mutation in TP53. Among NSCLC with non-ACA histology, 3.8% (6/156) of patients were found to have had a MET exon 14 skipping event—largely those in the SC group (21.1%, 4/19). Furthermore, 75.0% (3/4) of those had a concomitant mutation in TP53. The remaining two tumors had SCC histology. One of the tumors displayed a correlated TP53 mutation. Among advanced-stage ACA, evidence of an MET exon 14 skipping event was found in 2.6% (17/650) of patients, and 58.8% (10/17) of these patients had a concomitant mutation in TP53. Tumors with non-ACA histology showed MET exon 14 skipping in 2.8% (5/177) of cases. Again, the majority of tumors were in the SC group (18.2%, 4/22), but one additional patient showed ASC histology. Two out of four (50%) patients with SC also had a TP53 mutation. The ASC tumor had no additional mutations that we detected. 

### 2.11. Other Mutations and Fusions

The most common alterations in both early stages and advanced stages of NSCLC were mutations in the tumor suppressor gene TP53 (46.9%, 339/723 and 51.9%, 429/827, respectively). Regarding tumors with non-ACA histology, we found significantly higher rates of TP53 mutations than in ACAs: 71.8% (112/156) vs. 40.0% (227/567) (*p* < 0.001) in early-stage tumors and 67.2% (119/177) vs. 47.7% (310/650) (*p* < 0.001) in advanced-stage tumors. However, these passenger mutations often occur as co-mutations in association with driver alterations (e.g., EGFR, KRAS, BRAF, ERBB2, ALK, ROS, and RET). Nevertheless, in 14.8% (84/567) of ACAs and in 41.7% (65/156) of tumors with non-ACA histology in early stages and in 16.2% (105/650) of ACAs and 39.5% (70/177) of tumors with non-ACA histology in advanced stages, TP53 mutations were also found alone—that is, without associated driver mutations. Mutations in FGFR1, FGFR2, FGFR3, and FGFR4 genes dominated in the group of patients with non-ACA histology compared with the ACA group: 7.1% (11/156) in early stages and 4.0% (7/177) in advanced stages vs. 1.9% (11/567) in early stages and 2.3% (15/650) in advanced stages, respectively. Patients with SCC most frequently harbored FGFR mutations: 9.2% (8/87) in early-stage disease and 6.6% (7/106) in advanced-stage disease; mutations in FGFR1 (37.5%9) and FGFR3 (37.5%) were predominantly found in the early stage tumors, whereas mutations in FGFR3 (57.1%) and FGFR4 (28.6%) were most common in advanced-stage tumors. Two tumors, an ACA and an SCC, presented a fusion, both with the binding partner TACC3 (FGFR3–TACC3, F17T8 ACA, and F17T11 SCC). The distribution and frequencies of other mutations (STK11, PIK3CA, CTNNB1, PTEN, NRAS, MET, MAP2K, and ALK) occurring single or as concurrent mutations with other driver mutations are shown in Figure 4.

### 2.12. Programmed Death Cell-Ligand 1 (PD-L1) Expression

In total, 46.3% (321/693) of all NSCLC in early stage tumors (IA 93/236; IB 38/84, IIA 15/35, IIB 62/128, IIIA 113/210) and 53.0% (422/796) in advanced-stage tumors (IIIB 78/ 127, IIIC 21/37, IV 324/632) presented a positive expression of PD-L1 (TPS ≥1). Among them, 17.9% (124/693) versus 24.2% (193/796) showed high expression with a TPS score of ≥50%. Interestingly, significantly more tumors with negative PD-L1 expression (TPS = 0%) were found in early-stage tumors than in advanced-stage tumors (53.7%, 372/693 vs. 47.0%, 374/796, *p* = 0.0116). Moreover, tumors in advanced stages were significantly more likely to have high PD-L1 expression (TPS ≥ 50%) than tumors in early stages (24.2%, 193/796 vs. 17.9%, 124/693, *p* = 0.004), as shown in Figure 5.

Positive PD-L1 status was also detected in 25.7% of early-stage EGFR-mutated NSCLC, including 7.1% with high expression (TPS ≥ 50%). Of seven early-stage patients with ALK fusions, two had PD-L1 expression (TPS ≥1% < 50%). Tumors with high PD-L1 expression were not found. Patients with KRAS (p.G12C) mutations had PD-L1 expression in 58.8% of cases, including 24.5% with a TPS ≥ 50%. It should be pointed out that 25.5% (26/102) of KRAS-G12C-mutated patients with PD-L1 scores of ≥50 also had a functional TP53 mutation, which is important for treatment decision making. Of particular note is that there were also patients with an MET exon 14 skipping event in early stage tumors, and 52.6% of these patients had PD-L1 expression, including 26.3% with a TPS ≥ 50%. In advanced-stage tumors, the rates were significantly higher: 95.5% positive PD-L1 expression and 68.2% with a TPS ≥ 50%. Concurrent PD-L1 expression in tumors with ERBB2 mutations and ALK, ROS1, and RET fusions must be evaluated with caution due to small case numbers (Table 4). Detailed information on specific mutation frequencies in stage specific subgroups is presented in Appendix A. Due to low patient numbers per mutation and stage subgroup, no valid trends could be concluded.

## 3. Discussion

The aim of this study was to determine the prevalences and types of genomic alterations and PD-L1 expression in early-stage NSCLC (stages I–IIIA) in comparison to advanced NSCLC (stages IIIB–IV). Note that 46.9% of patients were diagnosed with early-stage disease, and 31.0% of tumors in the entire NSCLC cohort were stage IB–IIIA, potentially requiring adjuvant therapy. Next, 86.7% of the tumors in early stages showed one or more genomic alteration, and 37.0% of these tumors had genomic driver alterations for which approved therapeutic options already exist for advanced NSCLC. The prevalence of genomic driver alterations in early--stage tumors was comparable to that in advanced-stage tumors. Of note, EGFR-mutated ACAs in stage I–IIIA had significantly fewer co-occurring mutations than ACAs in the advanced stages (52.8% vs. 73.1%, *p* = 0.016), and TP53 was the most frequent mutation. This may have implications for prognosis and response to treatment. There was also a trend towards lower incidence of EGFR compound mutations in early-stage ACA (8.3% vs. 11.5% in advanced stages; *p* = 0.7991). Furthermore, 46.3% of early stage NSCLC patients were positive for PD-L1 (TPS ≥ 1%). Patients with driver mutations showed PD-L1 expression 58.8% of the time, and 24.5% had a TPS ≥ 50%.

Our study demonstrates the complex genomic landscape of biomarkers for targeted therapies and immunotherapies in early-stage NSCLC. Generally, NGS testing already allows identifying patients with EGFR mutations for whom a specific adjuvant TKI has been approved, but also provides a broad range of targets and predictive markers, giving the potential for exploring existing and new TKIs and CPIs in early-stage NSCLC.

In contrast to stage IIIB–IV NSCLC, in which precision oncology is now considered as state-of-the-art and molecular analysis as part of standard diagnostics, early-stage tumors are at present still rarely sequenced, despite the evident high recurrence rates of completely resected NSCLC. Consequently, there are limited data on the frequencies and distribution of EGFR mutations. In our institute, NGS reflex testing for oncogenic drivers and immunohistochemistry for PD-L1 status in all primary pulmonary ACA have been performed since 04/2017, and likewise in all other NSCLC cases with non-ACA histology since 11/2019. In the present study, a total of 2066 patients with an initial diagnosis of NSCLC were screened; complete TNM staging is available for 99.3% of patients. While advanced NSCLC was seen at the time of diagnosis in most patients, one-third presented with resectable tumors potentially requiring adjuvant therapy. The patient population with early-stage tumors did not differ significantly from the patients with the advanced-stage tumors with respect to gender distribution, age, smoking habits, and histological subtypes. However, patients with early-stage tumors exhibited significantly better histological grading compared to advanced-stage NSCLC patients (moderately differentiated G2 tumors: 53.7% vs. 24.5%, *p* < 0.001; poorly differentiated G3 tumors: 41.3% vs. 70.8%, *p* < 0.001). Of note, we detected lower rates of synchronous or metachronous secondary tumors in advanced NSCLC patients (1.2% vs. 5.7% in early-stage NSCLC, *p* < 0.001), which may be explained by the fact that in the advanced-tumor-stage patients, additional lung nodules were often clinically interpreted as metastases and therefore not examined histologically.

The proportion of SCC was lower in our study because we included NSCLC with non-ACAs in the reflex testing not before 11/2019. However, while some evidence indicated an increased proportion of EGFR mutations in younger patients and never-smokers in particular [43], we detected EGFR mutations only in one patient per cohort (1.2% and 1.1% in early and advanced NSCLC, respectively). 

The EGFR mutation rate in early-stage ACA patients was 12.7%, which is comparable to the mutation rate reported in the Caucasian patient cohort. The EGFR mutation rate in stage IIIB–IV ACA was similar (12.0%; *p* = 0.778). Most frequently, exon 19 deletions (45.8%) and *p*. L858R substitutions (26.4%) were detected. The double-blind, randomized ADAURA trial demonstrated that in patients with resected stage IB–IIIA NSCLC showing these specific mutations, DFS was significantly prolonged when receiving adjuvant osimertinib compared to the placebo. Osimertinib reduced the relative risk of disease recurrence or death by 79% in this study (DFS: HR 0.21, 95% CI: 0.16; 0.28; *p* < 0.0001). Patients receiving osimertinib had lower rates of loco-regional and distant metastases and less prognosis-determining brain metastases. The drug was administered over a 3-year period. Only mild TKI-specific side effects were observed. OS data have not yet been published [9]. Additional phase 2 trials [7,44] and phase 3 trials [8] evaluated the effect of TKI therapy in the adjuvant treatment of completely resected NSCLC with EGFR mutations. Despite significantly improved DFS, prolonged OS could not be demonstrated. If OS is not significantly prolonged by TKI treatment in completely resected early-stage NSCLC, the efficacy and harms of adjuvant therapy should be critically weighed against those of subsequent systemic therapies for the recurrence of disease.

To date, there are limited data on the specific types of EGFR mutations and co-mutations in early-stage NSCLC [22,35,36,37,38,39,40]. Studies of NSCLC in advanced stages (IIIB–IV) have shown that both the type of mutation (common or rare mutations, compound mutations, primary resistant mutations) and co-occurring mutations have an impact on prognosis and response to TKI therapy. For example, tumors with deletions in exon 19 revealed a more aggressive phenotype and a higher risk of distant metastases compared to tumors with L858R substitutions, particularly relating to brain metastases [22]. Moreover, a prospective trial demonstrated that the second-generation EGFR-TKI afatinib improved OS in patients with deletions in exon 19 but not in those with p.L858R EGFR mutations, providing a rationale for selecting patients who may actually benefit more from adjuvant EGFR TKI therapy. Uncommon mutations and compound mutations were reported to respond worse or not at all to TKI therapy in advanced NSCLC. Clinical benefit was also lower for patients with primary resistance-mediating EGFR mutations (e.g., p.T790M, EGFR exon 20 insertion) [45]. Therefore, it is important for treatment decisions to determine the type of EGFR mutation and utilize the information for treatment decisions. Generally, our study showed a similar distribution of EGFR mutation subtypes in both early and advanced NSCLC cohorts. However, we saw a trend towards a lower frequency of compound mutations in stages I–IIIA.

In terms of prognosis and response to treatment, the determination of passenger mutations that occur in addition to activating driver mutations has recently gained considerable scientific interest. Approximately 30% to 60% of EGFR-mutated ACA bear a concomitant mutation in the tumor-suppressor gene TP53. In a meta-analysis of 15 studies, Qin et al. demonstrated that corresponding TP53 mutations are associated with poorer progression-free survival (PFS) and OS. Specifically, patients with first-line treatment with EGFR TKIs had a significantly worse prognosis when concurrent TP53 mutations were present. The authors postulated that corresponding TP53 mutations caused primary TKI resistance in EGFR-mutated patients [46]. In our study, 34.7% (25/73) of patients in early tumor stages and 46.2% (36/78) of patients in advanced tumor stages had mutations in TP53 in addition to activating EGFR mutations. Canale et al. reported that patients with an exon 19 deletion had significantly shorter PFS and OS in the presence of an additional deletion mutation in exon 8 of TP53 [19]. Further, 6.9% (5/72) of all EGFR-mutated early-stage NSCLC patients in of our study revealed a combination of an exon 19 deletion and a corresponding TP53 mutation in exon 8.

In a cfDNA analysis of 1122 late-stage patients with EGFR-mutated ACA, Blakely et al. demonstrated that most patients (92.8%, 1043/1122) had at least one additional variant with known or probable functional significance in several other genes (TP53, PIK3CA, BRAF, MET, NF1, ERBB2, MYC, CDK6, and CTNNB1) in addition to the EGFR driver mutations. Relating to clinical outcomes, patients who responded to TKI therapy had significantly fewer additional genomic alterations than patients who did not respond. Other findings included that patients with PIK3CA gene alterations were less likely to respond to EGFR TKI therapy [47]. In our study, we detected activating co-mutations in the PIK3CA gene in 2.8% (2/72) of EGFR-mutated early-stage NSCLC patients. Additional driver mutations were found in SMAD4, CTNNB1, STK11, and BRAF. The extent to which the described co-occurring mutations affect prognosis and potential therapy with TKIs at early stages needs to be proven by further studies.

Beyond EGFR mutations, our study provides a broad overview of other driving alterations in early-stage NSCLC. Of particular importance are tumors with gene rearrangements. There are several trials currently enrolling that focus on a multimodality approach. These trials focus on patients with gene rearrangements in ALK, ROS1, NTRK1/2/3, and RET, but also BRAF V600 mutations and PD-L1 expression. In patients with ALK gene rearrangements, concurrent mutations in TP53 are of high interest, as these result in significantly lower median PFS and OS compared with TP53 wild-type tumors [48]. In our study, the rate of co-occurring mutations in TP53 was significantly lower in early tumor stages than in advanced tumor stages, although the analysis should be considered with caution due to the low detected frequencies of ALK gene rearrangements of 1.2% (7/567) and 2.2% (14/650) in the groups of early and advanced NSCLC, respectively.

While there are encouraging approaches with the ADAURA trial and other ongoing trials on targeted therapies for NSCLC with adjuvants and neoadjuvants, for the majority of patients without evidence of driver alterations, treatment options have not improved for more than 17 years since the publication of the landmark International Adjuvant Lung Cancer Trial study showing the efficacy of adjuvant chemotherapy [4]. The results of the IMpower010 trial have recently provided another promising treatment option showing a DFS benefit with atezolizumab compared with the best supportive care after adjuvant chemotherapy in patients with resected stage II–IIIA NSCLC. A pronounced benefit was noted in the subgroup of PD-L1-positive tumors (TPS ≥ 1%) [27]. Considering that 50.9% (190/373) of all stage II–IIIA tumors assessed in our study had PD-L1 expression of ≥1%, a large group of patients may potentially benefit from this new therapeutic option. Data from other randomized phase 3 trials exploring adjuvant treatment with PD-L1 and PD-1 inhibitors will provide further insights into the effect of immunotherapy in adjuvant treatment [26,27,28,29]. Several phase 2 trials already reported promising efficacy and safety for neoadjuvant PD-L1 and PD-1 inhibitors (LCMC3, NEOSTAR, NADIM) [49,50,51], and phase 3 trials are under-way (KEYNOTE-671, Checkmate 816, IMpower 030) [28,52,53]. In the first published interim analysis in the phase III Checkmate 816 trial, patients with resectable NSCLC and neoadjuvant nivolumab and chemotherapy had significantly higher rates of event-free survival and pathological complete response compared to the controls receiving neoadjuvant chemotherapy only. However, no significant difference was seen relating to overall survival [28]. Interestingly, in the multicentric, phase-II, single-arm NADIM trial with potentially resectable stage IIIA patients receiving neoadjuvant nivolumab plus carboplatin/paclitaxel, ctDNA levels were predictive for overall survival and RECIST v1.1 criteria were not [49].

NSCLC tumors with driver mutations in addition to PD-L1 expression are of particular interest and need to be more thoroughly investigated in future studies, as these tumors are currently mostly excluded from clinical trials. Due to a lack of evidence, it remains unknown whether the poor response rates to targeted therapies seen in advanced NSCLC patients also apply to early-stage disease and whether other therapeutic strategies are needed to address this issue. We detected positive PD-L1 status in as much as 25.7% of early-stage EGFR-mutated NSCLC patients, including 7.1% with high PD-L1 expression (TPS ≥ 50%). An ongoing single-arm, open-label, single-center phase-II trial aims to evaluate the clinical feasibility and safety of neoadjuvant sintilimab plus chemotherapy in patients with EGFR-mutated stage IIB–IIIB NSCLC (NEOTIDE, NCT05244213). Regarding other relevant molecular alterations in our study, patients with KRAS (p.G12C) mutations had positive PD-L1 expression in 58.8% of cases, including 24.5% with a TPS ≥ 50%. Moreover, 25.5% of these patients also had a functional TP53 mutation. Frost et al. showed that patients with the KRAS mutation G12C, high PD-L1 expression on tumor cells, and functional TP53 mutations are long-term responders to first-line palliative treatment with pembrolizumab monotherapy in advanced tumor stages [54]. In addition, of note, patients with a MET exon 14 skipping event had positive PD-L1 expression 52.6% of the time, including 26.3% with PD-L1 expression of TPS ≥ 50%. The extent to which these patients may benefit from targeted or immunotherapies remains to be investigated.

## 4. Materials and Methods

### 4.1. Patients

From April 2017 to February 2021, 2066 patients were primarily diagnosed with NSCLC in the Lung Cancer Center, HKEvB, Berlin, Germany. The Lung Cancer Service of HKEvB has been annually certified by the German Cancer Organization since 2009. The applied diagnostic and therapeutic algorithms have been based on the recommendations of the national German lung cancer guideline since its first publication in 2010 and its subsequent updates [55]. Histological diagnosis was performed by experienced pathologists in accordance with WHO criteria using the four-eye principle. In surgically resected NSCLC patients, the pathological tumor stages were used within this study, which were assigned according to the American Joint Committee on Cancer and International Union for Cancer Control (UICC) TNM staging system for lung cancer (8th edition). For all other NSCLC patients, the clinical UICC stages at the time of first diagnosis were utilized. Tumor tissue from 1550 patients was available for molecular pathologic analysis. In the remaining 516 patients, molecular testing was not performed because either a sufficient quantity of tissue could not be obtained and liquid biopsy yielded a negative result or DNA was of poor quality. Daily routine data from the indicated period were retrospectively analyzed for the study. Genomic analysis of tumor tissue was approved by all participating institutions, and informed consent was obtained from all subjects involved in the study. The research protocols of two studies also involving genomic analysis of tumor tissue from these patients were reviewed and approved by the Ethics Committee (Eth-X-AD/19 and Eth-48/20) of the Berlin Medical Association.

### 4.2. Nucleic Acid Extraction and Quantification

Tumor tissue was fixed with 4% buffered formalin and embedded in paraffin. Two 20-μm-thick unstained paraffin sections were prepared from each tumor, followed by an HE section to estimate the percentage of the tumor cell content. The tumor tissue was dissected using light microscopy and scraped from the unstained paraffin section. Nucleic acid extraction was automated using the Promega Maxwell 16 FFPE PLUS LEV DNA Kit or RNA Kit on the Maxwell instrument (Promega, Madison, WI, USA) following the manufacturer’s instructions. The concentration of the extracted nucleic acids was determined using the QuBit^®^ dsDNA HS Assay Kit or Qubit^®^ RNA HS Assay Kit in the Qubit 3.0. fluorometer from Invitrogen (Invitrogen/Thermo Fisher Scientific, Waltham, MA, USA). DNA quantification was then performed using the TaqMan^®^ RNAse P assay. The cDNA synthesis from the extracted RNA was performed by a Superscript VILO cDNA Synthesis Kit (Invitrogen, Thermo Fisher Scientific, Waltham, MA, USA) in the SimpliAmp Thermalcycler (Thermo Fisher Scientific, Waltham, MA, USA).

### 4.3. Next-Generation Sequencing

As of April/2017, mutation status was determined by NGS panel sequencing using the DNA Community Panel CLv2 (AKT1, ALK, BRAF, CTNNB1, DDR2, EGFR, ERBB2, ERBB4, FBXW7, FGFR1, FGFR2, FGFR3, KRAS, MAP2K1, MET, NOTCH1, NRAS, PIK3CA, PTEN, SMAD4, STK11, TP53) for 1078 patients in total (531 of early stages, 547 of advanced stages), and for the remaining 147 patients (39 of early stages and 108 of advanced stages) as of March/2020, using the nNGMv2 lung panel (ALK, BRAF, CTNNB1, EGFR, ERBB2, FGFR1, FGFR2, FGFR3, FGFR4, HRAS, IDH1, IDH2, KEAP1, KRAS, MAP2K1, MET, NRAS, NTRK1, NTRK2, NTRK3, PIK3CA, PTEN, RET, ROS1, STK11, TP53) by performing amplicon-based next-generation sequencing on the Ion Torrent S5 XL and Ion Torrent S5 Prime sequencing platforms (both Ion Torrent by Thermo Fisher Scientific, Waltham, MA, USA). In order to summarize the results of both panels used, only genes present in both panels were evaluated: ALK, BRAF, CTNNB1, EGFR, ERBB2, FGFR1, FGFR2, FGFR3, KRAS, MAP2K, MET, NRAS, PIK3CA, PTEN, STK11, TP53. Fusion analysis was also performed on the Ion Torrent S5 XL and Ion Torrent S5 Prime sequencing platforms (both Ion Torrent by Thermo Fisher Scientific, Waltham, MA, USA) using the amplicon-based Oncomine Focus RNA assay, which comprises 284 different fusion transcripts, including the genes ALK, AXL, BRAF, EGFR, FGFR1, FGFR2, FGFR3, MET, NTRK1, NTRK2, NTRK3, PAX8, RAF1, RET, ROS1, TMPRSS2, and others (for the complete list of all 284 fusions, see Appendix A).

### 4.4. Bioinformatics

For DNA coverage analysis and variant calling, bioinformatic pipelines in Torrent Suite 5.12, OS Ubuntu 14.04, was applied, and variant classification was performed with annovar. RNA sequences were uploaded from Torrent Suite into IonReporter Server 5.10, and RNA fusion analysis was performed on the latter. All data were electronically transferred on to a laboratory information system (ionLIMS, Heidelberg, Germany).

### 4.5. Immunohistochemistry

PD-L1 immunoreactions were visualized on 3 µm sections of FFPE tumor samples that were cut and mounted on Superfrost™ Plus Adhesion Microscope Slides (Epredia, Netherlands B.V.) by using the BOND™ Ready-to-Use Primary Antibody Programmed Death Ligand 1 (73–10), catalog number PA0832, on the automated BOND system in combination with BOND Polymer Refine Red Detection (Leica Biosystems Newcastle Ltd., Balliol Business Park, Benton Lane, Newcastle Upon Tyne NE12 8EW, UK), following the manufacturer’s instructions. PD-L1 protein expression on tumor cells was determined according to the tumor proportion score (TPS). No or ≤1% partial or complete membrane staining at each intensity was scored as 0, 1–49% was scored as 1+, and ≥50% was scored as 3—high PD-L1 expression. For quality assurance of the antibody, we participate annually in the round robin of QUIP (Quality Assurance Initiative Pathology GmbH, audited by the European Society of Pathology).

### 4.6. Statistical Analysis

Descriptive statistics were used for all variables of interest. Categorical variables were presented as frequencies and percentage values. Differences between variables were evaluated by using the Pearson’s Chi-square test. A *p*-value of *p* < 0.05 was considered statistically significant. All statistical analyses were conducted with RStudio 2022.07.1, RStudio, Boston, MA, USA.

## 5. Conclusions

In summary, this study provides a comprehensive overview of the genomic landscape of NSCLC in early resectable stages I–IIIA. For many of the detected biomarkers, approved therapeutic options are already available depending on evidence of genomic driver alteration and PD-L1 status. Given the complexity of genomic alterations and the immunogenic microenvironment, we strongly emphasize the need for comprehensive molecular reflex testing of all primary diagnosed non-small-cell lung cancers regardless of tumor stage to realize timely personalized therapy.

## Figures and Tables

**Figure 1 ijms-23-12511-f001:**
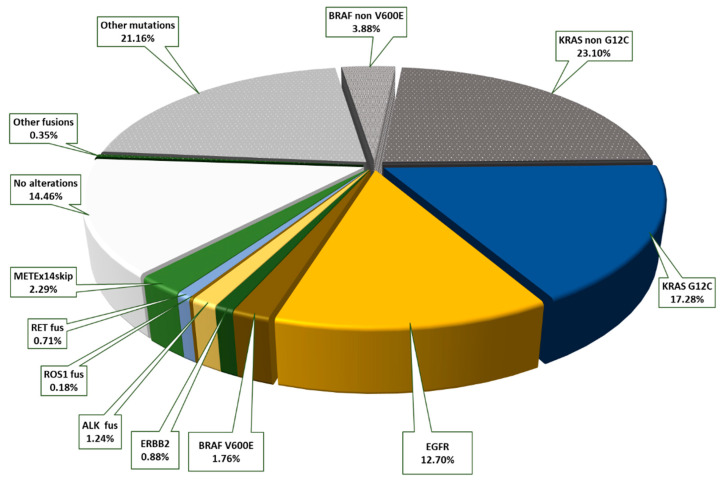
Distribution of genomic alterations in early-stage (I–IIIA) lung adenocarcinomas with the proportions of mutations and gene rearrangements for which there are currently approved targeted therapies highlighted in color (37.0% of all patients overall); the proportion of additional genomic alterations without a current targeted treatment option is outlined in gray (48.5% of all patients), and tumors without genomic alterations are shown in white (14.5% of all patients).

**Figure 2 ijms-23-12511-f002:**
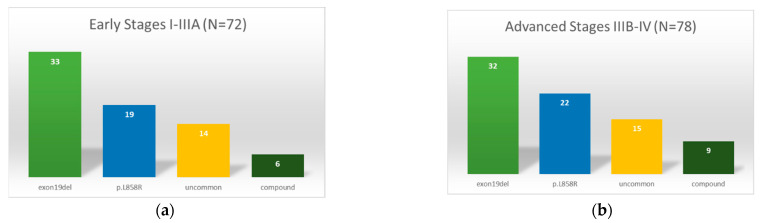
Classification and distribution of EGFR mutations in pulmonary adenocarcinoma: (**a**) adenocarcinoma in early-stage NSCLC (stages I–IIIA), (**b**) adenocarcinoma in advanced NSCLC (stages IIIB–IV).

**Figure 3 ijms-23-12511-f003:**
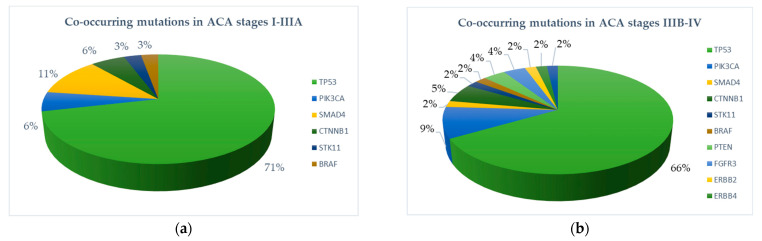
Proportions of co-occurring mutations in EGFR-mutated primary pulmonary adenocarcinoma. (**a**) Early-stage ACA (stages I–IIIA). (**b**) Advanced-stage ACA (stages IIIB–IV).

**Figure 4 ijms-23-12511-f004:**
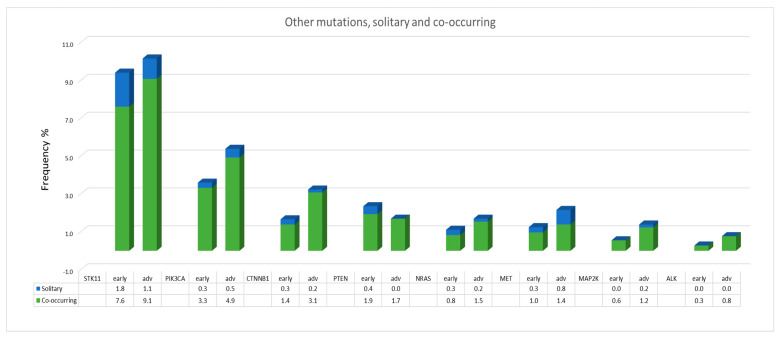
Frequencies of detected single and co-occurring mutations.

**Figure 5 ijms-23-12511-f005:**
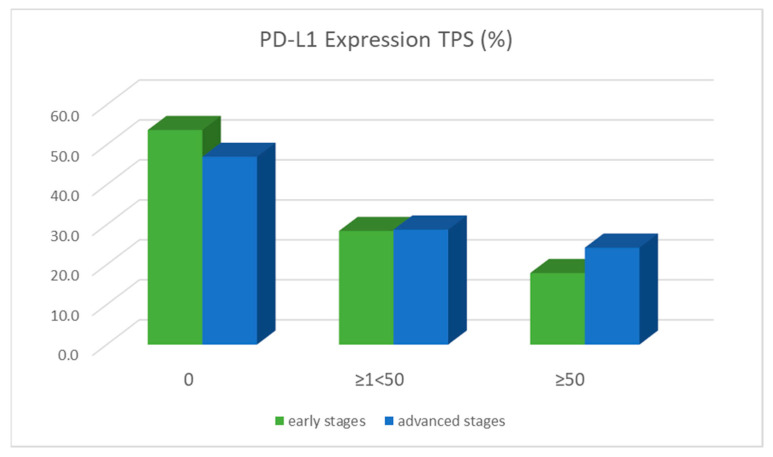
PD-L1 expression in early versus advanced tumor stages regarding the percentage of membrane-stained tumor cells (TPS score).

**Table 1 ijms-23-12511-t001:** Baseline patient characteristics.

	Early-Stage NSCLC (Stages I–IIIA)	Advanced-Stage NSCLC (Stages IIIB–IV)	*p*-Value
	N (%)	N (%)	
**No. of patients**	723 (46.6%)	827 (53.4%)	
**Age**			
≤60 years	157 (21.7%)	180 (21.8%)	1
>60 years	566 (78.3%)	647 (78.2%)	1
**Gender**			
Men	353 (48.8%)	445 (53.8%)	0.0564
Women	370 (51.2%)	382 (46.2%)	0.0564
**Histological NSCLC subtypes**			
Adenocarcinoma	567 (78.4%)	650 (78.6%)	0.9830
Squamous Cell Carcinoma	87 (12%)	106 (12.8%)	0.6970
Adenosquamous Carcinoma	15 (2.1%)	13 (1.6%)	0.5822
Sarcomatoid Carcinoma	19 (2.6%)	22 (2.7%)	1
Large Cell Carcinoma	14 (1.9%)	2 (0.2%)	0.0015
Large Cell Neuroendocrine Carcinoma	11 (1.5%)	8 (1%)	0.4487
Other NSCLC subtypes	1 (0.1%)	4 (0.5%)	0.3798
NSCLC not otherwise specified (NOS)	9 (1.2%)	22 (2.7%)	0.0713
**Stage according to UICC (8th edition)**			
IA1	43 (2.8%)		
IA2	126 (8.1 %)		
IA3	73 (4.7%)		
IB	89 (5.7%)		
IIA	36 (2.3%)		
IIB	135 (8.7%)		
IIIA	221 (14.3%)		
IIIB		134 (8.6%)	
IIIC		37 (2.4%)	
IVA		246 (15.9%)	
IVB		410 (26.5%)	
**WHO-Grading as available N (%)**	605 (83.7%)	534 (64.6%)	
G1	7 (1.2%)	1 (0.2%)	0.0736
G2	325 (53.7%)	131 (24.5%)	<0.001
G3	250 (41.3%)	378 (70.8%)	<0.001
G4	23 (3.8%)	24 (4.5%)	0.6619
**Smoker status as available N (%)**	432 (59.8%)	540 (65.3%)	
Never smoker	14 (3.2%)	30 (5.6%)	0.1165
Former smoker	162 (37.5%)	196 (36.3%)	0.7492
Current smoker	256 (59.3%)	314 (58.1%)	0.7764

**Table 2 ijms-23-12511-t002:** Comparison of the detection rates of actionable mutations in early-stage lung adenocarcinoma in primary tumors and loco-regional lymph-node metastases.

	Primary Tumor (n = 514)	Loco-Regional Lymph Node (n = 44)	*p*-Value
	N (%)	N (%)	
EGFR	68 (13.2%)	4 (9.1%)	0.6381
KRAS p.G12C	87 (16.9%)	10 (22.7%)	0.4429
BRAF p.V600E	10 (1.9%)	1 (2.3%)	0.5983
ERBB2	5 (1%)	0 (0%)	1
ALK	7 (1.4%)	0 (0%)	1
ROS-1	1 (0.2%)	0 (0%)	1
RET	5 (1%)	1 (2.3%)	0.3905
MET Exon 14 skipping	13 (2.5%)	3 (6.8%)	0.1243
No alterations	67 (13%)	7 (15.9%)	0.7581

**Table 3 ijms-23-12511-t003:** Frequencies of genomic alterations in lung adenocarcinoma, detected according to tumor stage.

	Early Stage (n = 567)	Advanced Stage (n = 650)	*p*-Value
	N (%)	N (%)	
EGFR	72 (12.7%)	78 (12%)	0.7777
KRAS G12C	98 (17.3%)	102 (15.7%)	0.5029
KRAS non G12C	131 (23.1%)	158 (24.3%)	0.6711
BRAF V600E	10 (1.8%)	14 (2.2%)	0.7782
BRAF non V600E	22 (3.9%)	19 (2.9%)	0.4450
ERBB2	5 (0.9%)	10 (1.5%)	0.4359
Other mutations	120 (21.2%)	159 (24.5%)	0.1947
ALK	7 (1.2%)	14 (2.2%)	0.3135
ROS1	1 (0.2%)	4 (0.6%)	0.3798
RET	4 (0.7%)	3 (0.5%)	0.7112
MET Exon 14 skipping	13 (2.3%)	17 (2.6%)	0.8597
Other fusions	2 (0.4%)	17 (2.6%)	0.0017
No alterations	82 (14.5%)	71 (10.8%)	0.0766

**Table 4 ijms-23-12511-t004:** Proportion of PD-L1-expressing tumor cells in early-stage vs. advanced-stage NSCLC regarding co-occurring actionable mutations and gene rearrangements.

		Stage I–IIIA	Stage IIIB–IV	*p*-Value
	PD-L1	N (%)	N (%)	
EGFR	TPS ≥ 1% < 50%	18/70 (25.7%)	25/75 (33.3%)	0.4112
	TPS ≥ 50%	5/70 (7.1%)	9/75 (12%)	0.4042
KRAS G12C	TPS ≥ 1% < 50%	60/102 (58.8%)	68/107 (63.6%)	0.5760
	TPS ≥ 50%	25/102 (24.5%)	38/107 (35.5%)	0.1136
BRAF V600E	TPS ≥ 1% < 50%	6/10 (60%)	14/16 (87.5%)	0.1627
	TPS ≥ 50%	2/10 (20%)	6/16 (42.9%)	0.4198
ERBB2	TPS ≥ 1% < 50%	2/5 (40%)	3/12 (25%)	
	TPS ≥ 50%	0/5 (0%)	1/12 (8.3%)	
ALK	TPS ≥ 1% < 50%	2/7 (28.6%)	5/14 (35.7%)	
	TPS ≥ 50%	0/7 (0%)	1/14 (7.1%)	
ROS1	TPS ≥ 1% < 50%	1/1 (100.0%)	2/4 (50%)	
	TPS ≥ 50%	0/0 (0%)	2/4 (50%)	
RET	TPS ≥ 1% < 50%	3/4 (75%)	3/4 (75%)	
	TPS ≥ 50%	1/4 (25%)	2/4 (50%)	
METExon14skip	TPS ≥ 1% < 50%	10/19 (52.6%)	21/22 (95.5%)	0.0024
	TPS ≥ 50%	5/19 (26.3%)	15/22 (68.2%)	0.0122

## Data Availability

The datasets analyzed in this study are available from the corresponding author upon request.

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
