# Peer review of "Landscape of Genomic Alterations and PD-L1 Expression in Early-Stage Non-Small-Cell Lung Cancer (NSCLC)—A Single Center, Retrospective Observational Study"

_ijms, 2022, doi:10.3390/ijms232012511_

Round 1
Reviewer 1 Report
In their study Stephan-Falkenau et al present an overview of the genomic landscape of early-stage NSCLC from a single center in Germany. Below are some suggestions to improve the manuscript.
1. The title needs to reflect that these results come from a single center.
2. The authors need to replace the term "genetic" with the term "genomic" throughout the text.
3. The results of trials using neoadjuvant immunotherapy in early-stage NSCLC (NADIM, CM 816) have been published. Updated results of the ADAURA study have also been presented in ESMO 2022. The authors need to include these in their Introduction and Discussion.
4. The fact that sampling site is not available for a substantial proportion of study participants is worrying. The authors need to state why this happened. Also, they need to discuss about additional limitations of this study, including the fact that all patients came from a single center.
5. Due to the small number of patients diagnosed from a lymph node biopsy, Table 2 can go to the Supplement.
6. The authors need to reduce the word count and be more to-the-point in the Discussion.
Author Response
Dear Reviewer,
After thorough revision of the manuscript, we are now sending you the current version with all comments on the changes made and after consulting the IJMS language revision service. On behalf of all the authors, we thank you for your review and the valuable and very helpful remarks to improve the manuscript. Below you will find the point-by-point analysis of all comments.
Point-by-point response 1st round review on ijms-1920623: Landscape of genomic alterations and PD-L1 expression in early stage non-small cell lung cancer (NSCLC) – a single center, retrospective observational study, Susann Stephan Falkenau et al.
Reviewer 1
In their study Stephan-Falkenau et al present an overview of the genomic landscape of early-stage NSCLC from a single center in Germany. Below are some suggestions to improve the manuscript. |
Thanks a lot for your very valuable and constructive suggestions for improvement! |
The title needs to reflect that these results come from a single center. |
Valid point, “– a single center, retrospective observational study“ has been added to the title. |
The authors need to replace the term "genetic" with the term "genomic" throughout the text. |
Valid point again, we have replaced “genetic“ by “genomic“ throughout the entire manuscript. |
The results of trials using neoadjuvant immunotherapy in early-stage NSCLC (NADIM, CM 816) have been published. Updated results of the ADAURA study have also been presented in ESMO 2022. The authors need to include these in their Introduction and Discussion. |
Thanks a lot for this worthwhile, we have added the specific references as well as a related discussion text section accordingly. Yet, we would like to refrain from refering to the recent ADAURA data (ESMO 2022, Paris) since peer-review process is pending. |
The fact that sampling site is not available for a substantial proportion of study participants is worrying. The authors need to state why this happened. Also, they need to discuss about additional limitations of this study, including the fact that all patients came from a single center. |
Thank you very much for this most helpful comment. Indeed, we made a mistake here: we confused at this point in the early tumor stages the representation of all NSCLC with the representation of ACA only. We have now added the sampling location for all NSCLC. In fact, the sampling location is known in more than 98% of cases. We haved moved the data for ACA only in early tumor stages down to Table 2. |
Due to the small number of patients diagnosed from a lymph node biopsy, Table 2 can go to the Supplement. |
We do see the point, but would like to keep table 2 in the main manuscript since this piece of information seems important to us. |
The authors need to reduce the word count and be more to-the-point in the Discussion. |
We have revised the document accordingly, attempting to bring the complexity of the findings into sharper focus in the discussion. In addition, we have sought greater linguistic clarity by consulting the IJMS language revision service. |
Reviewer 2 Report
Comment: Accepted after minor revision
The manuscript provides an important route for early stage NSCLC diagnosis and therapy by retrospectively analyze the both the ctDNA and PD-L1 status in 2066 patients. Combinational assay of genomic and proteomic is very novel and expected to have profound impact in the clinical practice and outcome in early stage NSCLC.
I recommend publication in IJMS in the current form, but if possible, including the following revision.
Minor Comments for consideration:
(1) In addition to ctDNA, the author might consider also include the miRNA marker panel for early stage NSCLC.
(2) Is it possible to also find out the related indeterminante pulmonary nodules (IPNs) information? Imaging is still an important way in the early stage NSCLC screen. The correlation between the genetic and proteomic and the IPNs information will expect to provide more precise diagnosis results.
Author Response
Dear Reviewer,
After thorough revision of the manuscript, we are now sending you the current version with all comments on the changes made and after consulting the IJMS language revision service. On behalf of all the authors, we thank you for your review and the valuable and very helpful remarks to improve the manuscript. Below you will find the point-by-point analysis of all comments.
Point-by-point response 1st round review on ijms-1920623: Landscape of genomic alterations and PD-L1 expression in early stage non-small cell lung cancer (NSCLC) – a single center, retrospective observational study, Susann Stephan Falkenau et al.
Reviewer 2
The manuscript provides an important route for early stage NSCLC diagnosis and therapy by retrospectively analyze the both the ctDNA and PD-L1 status in 2066 patients. Combinational assay of genomic and proteomic is very novel and expected to have profound impact in the clinical practice and outcome in early stage NSCLC. I recommend publication in IJMS in the current form, but if possible, including the following revision. |
Thanks a lot for your kind remarks as well as your very valuable and constructive suggestions for improvement! |
In addition to ctDNA, the author might consider also include the miRNA marker panel for early stage NSCLC. |
Many thanks for your comment, for the RNA-based fusion diagnostics the amplicon-based Oncomine Focus RNA Assay was used. We have added the major fusion partners included therein in section "5.3. Next Generation Sequencing" but left the list of 284 total fusion partners in Supplementary Table S3 as indicated due to space constraints. |
Is it possible to also find out the related indeterminante pulmonary nodules (IPNs) information? Imaging is still an important way in the early stage NSCLC screen. The correlation between the genetic and proteomic and the IPNs information will expect to provide more precise diagnosis results. |
Very worthwhile comment. We have generated a separate supplementary table S4 on the frequency of genomic alterations and PDL1 expression in pulmonary adenocarcinoma according to early tumour stages splitting into stage I subgroups. |
Round 2
Reviewer 1 Report
Accept